# Cognitive–Affective Dynamics of Political Attitude Polarization: EEG-Based Behavioral Evidence from a COVID-19 Vaccine Mandate Task

**DOI:** 10.3390/bs15081043

**Published:** 2025-08-01

**Authors:** Jing Li, Zhiwei Xu

**Affiliations:** 1School of Political Science and Public Administration, Wuhan University, Wuhan 430072, China; 2School of Business, Hubei University, Wuhan 430062, China; zwxu@whu.edu.cn

**Keywords:** EEG, political polarization, behavioral neuroscience, cognitive engagement, right prefrontal cortex

## Abstract

Political polarization in policy evaluations arises from identity-driven cognitive–affective dynamics, yet the neural mechanisms underlying the real-time processing of policy texts remain unexplored. This study bridges this gap by employing EEG to capture neurobehavioral responses during a COVID-19 vaccine mandate judgment task. The analysis of 70 politically stratified participants revealed significantly elevated gamma1 (30–50 Hz) activity in the right prefrontal cortex among policy supporters, reflecting enhanced attentional engagement and value integration. These topographically specific neural dissociations demonstrate how ideological alignment modulates cognitive–affective processing. Our findings establish EEG as a robust tool for quantifying implicit identity-driven evaluations, offering new pathways to decode polarization in contested policy contexts.

## 1. Introduction

Understanding how citizens form attitudes toward public policies is a long-standing question in political psychology ([9]; [33]). Traditional methods such as surveys and interviews have provided valuable insights, yet they often fail to capture the rapid and implicit psychological mechanisms that underlie political support or resistance ([24]). Recent advances in cognitive neuroscience, particularly electroencephalography (EEG), allow researchers to detect millisecond-level neural responses, revealing the automatic cognitive and emotional processes activated during political information processing ([11]).

This neurocognitive approach is especially relevant in polarized contexts. Government-mandated COVID-19 vaccination policies have become highly contentious, blending ideological identity, risk perception, and moral reasoning. According to motivated reasoning theory, individuals tend to engage more with information that confirms their existing beliefs while dismissing incongruent content ([23]). These tendencies are reflected not only in self-reports and behavior but also in neural activity, particularly in the brain regions responsible for conflict detection, attention allocation, and value-based decision-making ([18]).

Furthermore, construal level theory suggests that individuals evaluate collective benefits (e.g., public health) differently from personal costs (e.g., bodily autonomy), depending on the psychological distance of the issue ([36]). When such policies impose moral obligations and personal trade-offs simultaneously, they may elicit competing cognitive responses: rational self-interest ([5]) versus norm-based evaluations ([38]), with neurocognitive evidence for this conflict ([22]).

Although previous research has identified the psychological factors shaping policy attitudes—such as perceived fairness, risk, and benefit ([22])—few studies have examined how these evaluations unfold at the neural level. Some recent EEG studies have begun to explore political cognition more directly. For example, early ERP components have been linked to ideological congruence in voter intention ([11]). [44] ([44]) further demonstrated that selective exposure to political content produces confirmation bias patterns in EEG activity ([44]).

However, little is known about how individuals with opposing political stances process full policy texts differently at both the neural and behavioral levels. The present study addresses this gap by integrating EEG measures with a controlled policy judgment task to examine attentional bias, motivated attention, and emotion–cognition interaction during exposure to a contested vaccine mandate. Specifically, we explore how ideological alignment shapes observable behavioral divergence in neural and cognitive response patterns, reflected through high-frequency EEG activity. Gamma-band oscillations are known to be associated with sustained attention, emotional salience, and value-based evaluation, and may serve as a neurophysiological marker of political engagement ([11]).

While we did not predefine specific regions or frequency bands for hypothesis testing, these findings informed our general interest in how high-frequency activity might relate to ideological engagement in this context. By identifying these neural differences, the study contributes to a more nuanced understanding of political polarization and offers implications for more effective public policy communication.

## 2. Theoretical Background and Research Hypotheses

### 2.1. Political Psychology and Policy Reception

Political psychology provides a foundational framework for understanding how individuals interpret, evaluate, and emotionally respond to public policies ([27]). Rather than processing policy content in a neutral or deliberative fashion, people often rely on motivated reasoning—a psychological mechanism whereby information is selectively filtered to protect existing beliefs and ideological identities ([6]; [23]). This means that even when exposed to identical policy messages, individuals may arrive at contrasting conclusions, driven less by the facts themselves than by the motivational need to preserve internal consistency and group affiliation.

One outcome of this dynamic is the experience of cognitive dissonance—the psychological discomfort that arises when one encounters information that contradicts deeply held values or ideological commitments ([2]). Research has shown that individuals often resolve such dissonance not by revising their beliefs, but by engaging in rationalization, dismissal, or attentional bias, selectively attending to information that minimizes internal conflict ([13]; [14]). These regulatory strategies are particularly prevalent in the processing of politicized policies, where attitude defense becomes emotionally and socially reinforced.

Political identity, as distinct from general political preference, functions as a core aspect of self-concept for many individuals. It not only guides the content of political judgments but also shapes the intensity of emotional responses. Studies have shown that individuals with strong partisan identities exhibit heightened emotional reactivity and resistance to counter-attitudinal information, especially when policy content is perceived as identity-threatening ([19], [17]). At the neural level, such identity-protective cognition has been associated with enhanced activation in the anterior cingulate cortex and amygdala—regions implicated in conflict monitoring, emotional regulation, and motivational salience ([11]).

From a cognitive–affective interaction perspective, emotion and cognition are not distinct modules but interdependent processes that jointly shape political judgment. The embodied cognition framework posits that affective states serve as inputs to higher-order evaluations, modulating how individuals interpret ideologically salient stimuli ([28]). For example, policies framed in moral or existential terms can evoke strong emotional responses that subsequently bias reasoning and memory encoding. Neuroimaging studies suggest that the ventromedial prefrontal cortex (vmPFC)—a region responsible for integrating emotional valence, subjective value, and goal-directed decisions—plays a critical role in motivated attention and ideological evaluation ([17]).

In addition to the well-established frameworks of motivated reasoning and identity-protective cognition, several other theoretical perspectives may offer valuable insights into how individuals process politicized policy content. Reactance theory posits that individuals may experience psychological resistance when perceiving external threats to their autonomy, which can manifest as attitudinal opposition or disengagement. This is particularly relevant in the context of vaccine mandates, where public health imperatives may be interpreted as coercive ([22]). Social dominance orientation and authoritarianism, meanwhile, capture deep-seated preferences for hierarchy, order, and in-group conformity—traits that have been shown to shape receptivity to state-imposed mandates and collective action ([31]). Additionally, affective intelligence theory ([26]) highlights how discrete emotional states, such as anxiety or enthusiasm, dynamically influence attention and evaluative processing. These affective responses can alter the depth and direction of political information engagement, potentially accounting for the variation in neural reactivity.

Finally, framing effects—such as whether a policy is presented in moral versus economic terms, or in collectivist versus individualist language—have been shown to modulate both emotional salience and cognitive elaboration ([37]). While the present study primarily focuses on ideological alignment and affective salience, these alternative mechanisms may intersect and interact in shaping neural and behavioral responses. Future research may benefit from incorporating these dimensions to build a more integrative account of how political attitudes and emotional frameworks converge in real-time policy evaluation.

In sum, contemporary research demonstrates that political behavior is not solely the product of rational cost–benefit calculus. Instead, it reflects a dynamic cognitive–affective processing system, whereby emotional reactivity, attentional focus, and identity-congruent motivation interact to shape evaluative judgments. These insights highlight the importance of exploring not just what people think about policies, but how they neurologically and behaviorally engage with them, especially under ideologically polarized conditions.

### 2.2. EEG and Political Behavior

Electroencephalography (EEG) has become an essential tool in understanding the neural mechanisms underlying political behavior, offering a high temporal resolution to capture rapid cognitive and emotional responses to political stimuli ([11]; [18]). Research indicates that specific EEG frequency bands, particularly gamma oscillations, play a crucial role in processing emotionally charged and self-relevant political information ([32]). Gamma oscillations, typically ranging from 30 to 50 Hz, are associated with higher-order cognitive functions such as attention, memory, and emotional processing. Studies have shown that increased gamma activity in the frontal regions of the brain correlates with enhanced cognitive processing and emotional arousal in response to stimuli ([11]; [18]; [40]; [41]).

For instance, [40] ([40]) observed that cortical activity during the observation of political speeches was significantly higher in supporters compared to swing voters, indicating heightened engagement and a greater emotional response to the content. This finding suggests that emotional engagement may play a significant role in processing persuasive messages, including those in political communication contexts ([40]). Similarly, research by [41] ([41]) highlighted that the narrative structure of video commercials influenced EEG spectral dynamics, with an increased gamma power observed in the limbic system during emotionally engaging content. This further supports the idea that emotional engagement plays a key role in processing persuasive messages, which is equally relevant in the context of political behavior.

In political contexts, understanding the neural mechanisms underlying the reception of policy messages is crucial. EEG studies have provided insights into how individuals’ political identities and emotional responses influence their processing of political information. For example, research by [11] ([11]) demonstrated that early EEG responses to pre-electoral survey items reflected political attitudes and could predict voting behavior, suggesting that neural measures can complement traditional self-report methods in assessing political preferences. Additionally, studies like [42] ([42]) have highlighted the predictive power of EEG features in consumer behavior, which may provide useful insights into how cognitive and emotional responses to political messages might similarly be predicted by EEG patterns. By examining how different political groups process policy information at the neural level, researchers can gain a deeper understanding the cognitive and emotional mechanisms that drive political attitudes and behaviors ([42]).

### 2.3. Why Mandatory Vaccination Matters: Theoretical Justification and Hypotheses

The decision to focus on a COVID-19 vaccine mandate policy is grounded in both empirical urgency and theoretical richness. In some countries, particularly those with high levels of political polarization, mandatory COVID-19 vaccination emerged as a contentious and emotionally charged public health measure. It presented a value conflict scenario, forcing individuals to weigh collective health obligations against personal autonomy, thereby eliciting strong emotional and ideological reactions ([1]; [25]). The emotional salience and symbolic significance of vaccine mandates make them particularly suited for emotionally charged decision-making tasks in experimental settings.

In China—despite the country’s reputation for strong top-down pandemic control—vaccination policy has remained nominally voluntary, underscoring the socio-political sensitivity of enforced medical interventions even in centralized regimes. This highlights the real-world relevance and cross-cultural applicability of mandatory vaccination as a policy stimulus. Moreover, with the future threat of more virulent pathogens, legal mandates for vaccination may re-enter public discourse globally, making this issue both temporally urgent and policy-relevant in the long term ([12]).

Theoretically, this topic represents an ideal intersection between political psychology and affective neuroscience. The tension between institutional authority and bodily autonomy activates key psychological mechanisms such as motivated reasoning ([23]), cognitive dissonance ([2]), and identity-protective cognition ([17]). EEG, with its high temporal resolution, allows for the detection of rapid neural dynamics associated with these processes, offering a window into how political beliefs are embodied and neurologically filtered during exposure to contested policy content ([28]).

Drawing on this framework, we position mandatory vaccination judgment as a naturalistic behavioral task in which neural data can reveal how ideological alignment modulates information processing. Prior EEG research has demonstrated that political attitudes can be reflected in distinct patterns of neural activity, especially in frontal brain regions linked to attention allocation, emotional salience, and reward-based evaluation ([11]; [29]). These insights suggest that supporters and opponents of a vaccine mandate may exhibit divergent patterns of attentional engagement and value processing—detectable in real time through EEG.

**H_1_:** 
*We expect to observe significant differences in EEG power between the supporter and opponent groups during exposure to the COVID-19 vaccine mandate document. These differences are anticipated to reflect divergent levels of cognitive and emotional engagement with the policy content.*


**H_2_:** 
*We further hypothesize that the direction of neural activity will align with political attitudes: individuals who support the policy will exhibit greater EEG power, particularly in frequency bands associated with motivated attention and value congruence processing, while those who oppose the policy will show patterns indicative of resistance, affective conflict, or disengagement.*


## 3. Methods

### 3.1. Participant

A total of 70 participants were recruited for this study, consisting of 35 individuals who supported the government-mandated COVID-19 vaccination policy and 35 who opposed it. Participants were initially recruited through a questionnaire distributed to 110 young adults aged 20–25 years (M = 22.82, SD = 1.31), a demographic stage characterized by social integration and the development of emotional belonging ([16]). The age range was specifically chosen to ensure that the EEG data would not be confounded by age-related variability, focusing on a homogeneous age group that represents individuals who are newly entering the workforce. To ensure the participants’ suitability for a neuroscience experiment, inclusion criteria required that individuals had at least 6 months of continuous work experience, indicating social independence.

The sample was geographically diverse, with 52.9% of participants being non-local residents, thus enhancing the cultural heterogeneity of the sample. Participants were classified based on their political stance regarding the vaccine mandate. Using a pre-study questionnaire, 40 participants who expressed a neutral or ambiguous stance on the policy were excluded from the study. The remaining 70 participants, whose political attitudes were clearly defined, were equally divided into two groups: 35 individuals who strongly supported the government’s vaccine mandate and 35 who expressed clear opposition to the policy. This sample size is relatively robust by EEG research standards, especially for between-group comparisons, as many similar studies often include 20 to 30 participants per group ([21]; [43]).

The supporter group consisted of 19 females and 16 males, while the opponent group comprised 17 females and 18 males, ensuring a balanced gender distribution across both groups. This approach was designed to minimize the potential confounding effects of gender while ensuring that both groups were comparable in terms of their demographic characteristics.

This study followed an exploratory data-driven approach. While our theoretical background highlighted the relevance of cognitive–affective processing and suggested that high-frequency EEG activity may be involved, we did not pre-register specific regions of interest (ROIs) or frequency bands prior to data analysis. The post hoc focus on gamma-band activity and the right prefrontal cortex emerged from significant group-level patterns in the data, which are interpreted in light of prior literature on political cognition and emotional salience.

### 3.2. Experimental Procedure

In the experiment, participants were seated in a quiet laboratory room in front of a computer screen. They were instructed to read a policy document related to the government-mandated COVID-19 vaccine, which was displayed on the screen for 5 min. The policy document was carefully crafted to mimic the style of official national-level government communications, using AI technology to replicate the tone, structure, and language commonly found in authoritative policy documents (see Appendix A for the full text of the stimulus). This ensured that the content was realistic and consistent with how participants might engage with such documents in a real-world context.

Prior to the stimulus presentation, the Emotiv EPOC system automatically recorded a brief 30 s segment (15 s eyes open, 15 s eyes closed) as part of its built-in calibration. This segment was not used in the analysis. The actual EEG data were collected during a 5 min period in which participants read the policy document at their own pace. This duration was chosen based on standard practices in EEG research, where 5 min recordings are commonly used to ensure stable signals without inducing fatigue ([39]). It also matched the average time required to process the full document, allowing us to capture sustained, uninterrupted cognitive and emotional engagement with the stimulus. Extending the session further might have introduced noise due to fatigue or drifting attention, so the 5 min window struck a balance between ecological validity and data quality.

### 3.3. EEG Equipment

EEG signals were recorded continuously using the Emotiv EPOC+ system, which consists of 14 active electrodes and 2 reference electrodes, positioned according to the international 10–20 system (Figure 1). This system ensures standardized and reliable electrode placement across participants. The data were sampled at a rate of 256 Hz, with a bandwidth of 0.2–43 Hz, and a notch filter was applied to eliminate noise from frequencies between 50 and 60 Hz, ensuring the accuracy of the recorded signals.

The EEG signals were transmitted wirelessly to a secured USB receiver and captured using the Emotiv Pro 3.0 software on a Windows 10 PC. To ensure the quality of the data, real-time monitoring of the channel quality was performed, and only data with >95% channel quality were retained for analysis. Data were segmented into 2 s epochs to allow for detailed analysis of the EEG responses during the stimulus presentation.

### 3.4. EEG Preprocessing

Raw EEG data were preprocessed using EEGLAB2021 ([8]) in MATLAB2021b, utilizing a fully scripted pipeline to ensure consistency across participants. The data were initially imported from ‘*.edf’*’ format and converted into EEGLAB-compatible ‘*.set* ‘ files.

Continuous EEG signals were band-pass filtered between 1 and 100 Hz to remove slow drifts and high-frequency noise. A notch filter was applied between 48 and 52 Hz to suppress powerline interference. The data were then segmented into non-overlapping 2-s epochs. Channels exhibiting poor signal quality were manually inspected, and spherical spline interpolation was used to interpolate the channels when necessary. Epochs with extreme amplitude values (±100 μV) were excluded to mitigate residual noise and artifacts.

Independent component analysis (ICA) was performed on the cleaned datasets using the extended Infomax algorithm, with principal component analysis (PCA) set to the number of effective channels. Artifact components related to ocular movements, muscle activity, and environmental noise were identified through the visual inspection of component topographies and time courses. These components were manually rejected following standard guidelines. After ICA, all datasets were visually re-checked and re-referenced to the common average reference to reduce spatial bias.

### 3.5. Power Spectral Analysis

To examine the spectral dynamics associated with political decision-making, we conducted a power spectral density (PSD) analysis on all EEG data segments corresponding to the policy document viewing period under both experimental conditions (supporter vs. opponent groups). PSD quantifies the distribution of signal power across different frequency components, providing insights into the engagement of distinct neural processes such as attention, cognitive control, and affective arousal ([3]; [42]; [45]).

For each 2 s epoch, PSD was estimated using the fast Fourier transform (FFT) algorithm. Let $x\left[n\right]$ represent the discrete EEG time-series sampled at rate $F_{s}$ = 256 Hz, and N be the number of time points per epoch (here, N = 512). The FFT of $x\left[n\right]$ is defined as follows:(1)$$X\left[k\right]=\sum_{n=0}^{N-1}\;x\left[n\right]\cdot e^{-j2\pi kn/N}$$

The single-sided power spectral density was then computed as follows:(2)$$PSD\left(f_{k}\right)=\frac{2}{F_{s}N}\cdot |X\left[k\right]{|}^{2}\;\mathrm{for}\;k=0,1,\ldots ,\frac{N}{2}$$
where $f_{k}=\frac{kF_{s}}{N}$ denotes the frequency in Hz. This approach yields PSD estimates in units of ${\mathrm{uV}}^{2}/\mathrm{Hz}$. The analysis was performed for each channel and epoch independently, resulting in a 3D matrix (participants × channels × frequencies) for each condition.

To identify frequency bands and spatial regions involved in policy-related cognition, we aggregated power across canonical EEG frequency bands: delta (1–3 Hz), theta (4–7 Hz), alpha1 (8–10 Hz), alpha2 (10–13 Hz), beta1 (14–18 Hz), beta2 (18–30 Hz), gamma1 (30–50 Hz), and gamma2 (50–80 Hz). For each participant and frequency band, power values were averaged over all epochs and frequency bins, yielding a topographical distribution of spectral power.

Statistical comparisons between the supporter and opponent groups were performed using t-tests for each frequency bin across channels. For fine-grained frequency analysis (1–80 Hz), we tested for significant differences at each frequency point across the spectrum and applied the Benjamini–Hochberg false discovery rate (FDR) correction to control for multiple comparisons ([4]). In addition, scalp topographies were visualized to illustrate the spatial patterns of spectral differences across the cortex.

This full-spectrum, multi-channel analysis enabled us to characterize the neural signatures of participants’ attitudes toward the policy with both spectral specificity and spatial resolution.

## 4. Results

### 4.1. Group-Averaged Power Spectra Reveal Condition-Specific Neural Profiles Across Frontal and Temporal Channels

To examine how neural oscillatory activity varied as a function of political stance, we computed grand-averaged power spectral density (PSD) curves for each of the 14 EEG channels spanning bilateral frontal, temporal, and occipital regions. For each electrode, the mean PSD across participants was calculated separately for the supporter and opponent groups over the 1–80 Hz frequency range.

As shown in Figure 2, several electrodes—particularly those over the right frontal and frontotemporal areas such as AF4, F8, and FC6—exhibited visibly elevated spectral power in the supporter group compared to the opponent group. In contrast, left frontal channels (e.g., AF3, F7, F3) and posterior sites (e.g., O1, O2) showed largely overlapping spectral profiles between groups, with minimal or no consistent differences.

The observed enhancement in the supporter group was most pronounced within the higher frequency bands (especially beta and gamma ranges), although the overall effect was not uniformly significant across all channels. These topographical patterns suggest that right-lateralized prefrontal and frontotemporal regions are more strongly recruited in the supporter group, potentially reflecting mechanisms of motivational salience, attentional engagement, and value-based evaluation under ecologically valid conditions.

### 4.2. Topographic Dissociation in High-Frequency EEG During Policy Evaluation

We generated topographic maps of EEG power spectral density (PSD) for eight canonical frequency bands: delta (1–3 Hz), theta (4–7 Hz), alpha1 (8–10 Hz), alpha2 (10–13 Hz), beta1 (14–18 Hz), beta2 (18–30 Hz), gamma1 (30–50 Hz), and gamma2 (50–80 Hz). For each frequency range, we calculated group-averaged PSD, independent-samples t-values, and *p*-value maps across all channels, with the significance determined after applying false discovery rate (FDR) correction for multiple comparisons.

As shown in Figure 3, Figure 4, Figure 5 and Figure 6, most low- and mid-frequency bands (delta, theta, alpha1, alpha2, beta1, beta2, gamma2) exhibited no significant differences between the supporter and opponent groups. However, one high-frequency band demonstrated statistically reliable effects: in the gamma1 band (30–50 Hz), the PSD was significantly higher in the supporter group compared to the opponent group over the right frontal electrode AF4 and its neighboring sites (Figure 6). This effect was significant at *p* < 0.05 after FDR correction.

These results suggest that enhanced high-frequency activity in the right prefrontal cortex is selectively associated with support for the policy, potentially reflecting the engagement of neural mechanisms underlying reward evaluation, motivational salience, or decisional commitment in ecologically valid contexts.

### 4.3. Spectral Range-Specific Statistical Contrasts at the Right Prefrontal Cortex

To identify the specific frequency components that differentiate the supporter and opponent groups, we conducted independent-samples *t*-tests on power spectral density (PSD) values extracted from the right prefrontal electrode AF4. EEG data were sampled at 256 Hz and segmented into 2 s epochs, yielding 512 data points per epoch. We performed fast Fourier transform (FFT) on each epoch using $N_{\mathrm{FFT}}=512$, which provided a frequency resolution of 0.5 Hz ($\Delta f=F_{s}/N_{\mathrm{FFT}}$). This enabled us to perform statistical comparisons at each 0.5 Hz increment from 0 to 80 Hz.

As shown in Figure 7, the t- and *p*-value curves across frequency bins revealed statistically significant differences between the supporter and opponent groups. Notably, *p*-values fell below the *p* < 0.05 threshold between approximately 30 and 45 Hz, corresponding to the gamma1 band, where the supporter group showed consistently higher PSD than the opponent group. This effect was significant after applying false discovery rate (FDR) correction for multiple comparisons.

These results reinforce the hypothesis that higher-frequency activity in the right prefrontal cortex is selectively enhanced in the supporter group. The observed increase in gamma1 activity is consistent with the engagement of neural mechanisms involved in cognitive integration, reward valuation, and attentional commitment to goal-congruent stimuli under ecologically valid conditions.

Based on the statistical contrasts performed across frequency bands and brain regions, the results provide strong support for both hypotheses. H_1_ is confirmed, as we observed significant differences in gamma1 activity between the supporter and opponent groups, particularly in the right prefrontal cortex. H_2_ is also supported, as the increased neural activity in the gamma band is associated with the supporter group, potentially reflecting higher levels of attentional commitment and cognitive processing in response to the policy document.

### 4.4. No Global Spectral Differences Across Frequency Bands After Whole-Brain Averaging

To determine whether the observed spectral effects were spatially localized or indicative of global shifts in oscillatory activity, we averaged the PSD values across all 14 EEG channels for each participant and performed independent-samples *t*-tests for each canonical frequency band (Table 1). As detailed in the analysis, none of the eight frequency bands (delta through gamma2) showed significant differences between the supporter and opponent groups at the whole-scalp level (all *p* > 0.05).

These results confirm that the spectral enhancements observed at AF4 and its neighboring sites are topographically specific, rather than reflecting diffuse, global activation shifts. This supports the interpretation that high-frequency activity in the right prefrontal cortex plays a focal role in the supporter group.

## 5. Discussion

### 5.1. Interpretation of Findings

The present study provides compelling evidence that neural activity in response to a government-mandated COVID-19 vaccination policy varies significantly between individuals who support the policy and those who oppose it. Specifically, our results highlight a pronounced difference in gamma1 band power (30–50 Hz) between the supporter and opponent groups, with significantly higher gamma1 activity observed in the right prefrontal cortex (AF4) of the supporter group. This finding is of particular significance for understanding the cognitive and emotional processes underlying political attitudes.

The gamma1 band is associated with high-frequency neural oscillations, which are thought to reflect processes of cognitive integration, attentional focus, and reward evaluation ([10]). Elevated gamma activity in the right prefrontal cortex suggests that the supporter group engages in more focused and evaluative processing of the policy content. This may reflect the valuation of the policy as personally relevant, where higher cognitive resources are devoted to processing, interpreting, and integrating the policy’s implications. In contrast, the opponent group, whose gamma activity was significantly lower in the same region, may have engaged in more negative or resistant emotional and cognitive responses to the policy, possibly due to the policy’s incongruence with their pre-existing beliefs and values.

These findings align with established theories in cognitive processing and emotional evaluation of policy content. Theories of motivated reasoning suggest that individuals process information in a way that aligns with their pre-existing attitudes or beliefs ([23]). In the case of the supporter group, the increased gamma1 power in the right prefrontal cortex may reflect a more motivated and engaged processing style, where the policy is processed in a manner that reinforces their existing political stance. This aligns with dual-process models of cognition, where more emotionally and personally relevant information tends to recruit higher-order cognitive and affective processing ([34]).

Moreover, the significance of the right prefrontal cortex in political attitude alignment is consistent with existing literature on the role of lateralized brain regions in emotion regulation, attentional focus, and decision-making. Research suggests that the right frontal cortex plays a crucial role in processing self-relevant stimuli and managing emotional responses ([7]; [35]). The observed neural activation in the right prefrontal cortex (AF4) in the supporter group highlights the involvement of this brain region in attentional control and evaluation of goal-congruent stimuli, reinforcing the idea that right-lateralized brain activity is linked to more positive emotional engagement with self-relevant content.

In summary, the enhanced gamma1 activity in the right prefrontal cortex of the supporter group provides strong evidence for the role of neural mechanisms in the evaluation of politically relevant content. This neural activation is not only associated with cognitive processing but also with the emotional and motivational salience of the policy document. The findings suggest that political attitudes, particularly those that are strongly held, are underpinned by dynamic cognitive and emotional processing in the right frontal regions of the brain.

Although the observed difference in gamma1-band activity over the right prefrontal cortex (AF4) was not pre-specified as a region or frequency of interest, it emerged consistently across participants and aligns with established theoretical frameworks. Given the exploratory nature of this study, we did not adopt a pre-registered hypothesis regarding precise frequency bands or electrode sites. Nevertheless, gamma-band activity in frontal regions has been widely associated with sustained attention, affective salience, and value-based decision-making in prior EEG research ([11]; [40]; [41]). The current findings, although data-driven, are therefore not inconsistent with existing theoretical and empirical literature.

Furthermore, while our EEG system does not support full source localization, the observed topographical distribution of gamma-band activity over the right prefrontal scalp region (AF4) invites theoretical inferences grounded in prior literature. Gamma oscillations in frontal areas have been closely associated with the ventromedial prefrontal cortex (vmPFC), a region involved in integrating emotional valence, subjective value, and self-relevant decision-making ([20]). This aligns with our interpretation that supporters may experience policy content as value-congruent, thus engaging reward-based and affective integration processes. Additionally, the anterior cingulate cortex (ACC) has been implicated in ideological conflict monitoring and the detection of belief-incongruent stimuli ([30]). The enhanced gamma activity in supporters may thus also reflect reduced conflict or dissonance when processing ideologically aligned content. Although these interpretations remain speculative given our spatial resolution limits, they help contextualize our scalp-level findings within broader neural models of motivated reasoning and political affect.

### 5.2. Theoretical Contributions

This study makes a novel contribution to the field of political behavior by using EEG to examine the neural processes underlying real-time policy reception. While the study of political behavior has largely relied on traditional self-report measures and behavioral data, EEG offers a unique opportunity to capture the instantaneous neural responses to policy content, providing a more direct and objective measure of participants’ cognitive and emotional engagement. By recording EEG signals during the evaluation of a government-mandated policy, we were able to identify specific neural patterns that differentiate individuals based on their political stance, an approach that has not been extensively explored in existing political neuroscience research.

The study provides significant neurocognitive insights into the mechanisms of political polarization and attitudinal processing during policy exposure. While much of the existing literature on political attitudes has focused on the cognitive and emotional processes involved in political decision-making, fewer studies have explored how these processes manifest neurologically during real-time interactions with policy content. Our findings suggest that right prefrontal activity, particularly in the gamma1 band, plays a crucial role in the evaluation of policy content, with enhanced neural activity in individuals who support the policy. This neural activation likely reflects attentional commitment, cognitive integration, and reward evaluation, which are all essential processes when evaluating policy that aligns with one’s political identity.

Furthermore, this study advances the understanding of political polarization by linking differences in neural activity to individuals’ political stances. The gamma1 frequency band’s selective enhancement in the supporter group provides a neurocognitive marker for how individuals process and evaluate information that aligns with their attitudinal beliefs. This highlights the role of emotionally laden stimuli in shaping political beliefs and suggests that neural mechanisms of motivated reasoning—where individuals process information that supports their existing views more intensely—are also reflected in neural patterns during real-time policy exposure. In essence, this study demonstrates how neuroscientific methods can provide a deeper understanding of the mechanisms underpinning political polarization and ideological divisions.

Thus, by integrating EEG into the study of political behavior, this research offers valuable contributions to the neuroscience of political decision-making and attitude formation. It underscores the importance of examining neural responses to political stimuli, not only to understand how people engage with policy information but also to inform strategies for addressing polarization and improving public policy communication.

### 5.3. Practical Implications

Understanding the neural responses to political policies, particularly in the context of political division, has significant implications for political communication strategies. In times of heightened political polarization, where ideological divides often lead to entrenched views and resistance to change, this study offers valuable insights into how individuals engage with policy content on a neural level. By identifying the neural patterns that differentiate supporters and opponents of a policy, political communicators can tailor their messages to better resonate with the cognitive and emotional processes of different groups. For example, understanding that supporters of a policy exhibit enhanced gamma1 activity—likely reflecting positive cognitive processing and attentional focus—suggests that messages framed in a way that reinforce existing beliefs could increase engagement and support. On the other hand, the lower neural activity observed in the opponent group may indicate that political messages need to be restructured to address potential cognitive dissonance and resistance.

These findings also emphasize the role of emotional and cognitive engagement in policy communication. Politicians and communicators can improve the effectiveness of their messages by considering the emotional resonance of policy content, recognizing that emotionally laden messages may be more successful in engaging individuals’ cognitive and emotional responses. For instance, framing policies in a way that appeals to shared values or goals can activate the reward system and promote a sense of alignment, especially for individuals who might be initially opposed to the policy. This approach could help to mitigate cognitive dissonance, which occurs when individuals are confronted with information that conflicts with their pre-existing beliefs or values. By understanding how to frame policies in ways that are emotionally engaging and cognitively congruent with an individual’s political stance, political communicators can reduce resistance and increase overall public support.

For policymakers, these insights suggest several strategies for framing policies in a way that may promote greater public acceptance, even in polarized contexts. First, policymakers could consider emphasizing the personal relevance and emotional significance of policies to increase cognitive and emotional engagement. By framing policies as aligning with the core values of diverse groups, policymakers could reduce the perceived threat to individuals’ identity and increase acceptance. Second, addressing cognitive dissonance by presenting policies in a manner that acknowledges the concerns of opposing groups, while offering compelling and reassuring narratives, could reduce emotional resistance and foster greater support. Finally, providing clear, consistent information and transparent reasoning for policy decisions can help mitigate resistance by making the policy appear more justifiable and aligned with public interests.

In summary, the findings from this study suggest that understanding neural responses to political policies offers a unique opportunity to refine communication strategies in politically divided times. By tailoring messages to engage cognitive and emotional processes effectively, policymakers can foster greater public engagement, reduce resistance, and ultimately, enhance the effectiveness of their policies.

### 5.4. Limitations and Future Research

This study has several limitations. The sample size of 70 participants, while sufficient for detecting effects, limits the generalizability of the findings. Future research should include larger and more diverse samples to enhance robustness and representativeness. Additionally, the study focused on a specific age group (20–25 years), which may not capture neural responses across different age groups or socio-economic backgrounds.

The focus on a single political issue, COVID-19 vaccination, may limit the generalizability of the results to other policy contexts. Future studies should examine neural responses to a broader range of political issues to determine whether the observed neural patterns are specific to the COVID-19 policy or reflect more general cognitive and emotional processes related to political evaluation.

Additionally, although the policy document was AI-generated, it closely followed real government proposal language and addressed a highly salient public issue in China. Still, we acknowledge that using authentic or multimodal policy content (e.g., videos, social media) in future research would improve the ecological validity.

Further research should also explore cross-cultural comparisons to understand how cultural context influences neural processing of political content. Longitudinal studies are needed to track how neural responses to political policies evolve over time, providing insights into the dynamics of political attitudes. Finally, interventions like neurofeedback could help reduce polarization by modulating neural responses to political content, offering potential for fostering more constructive political engagement.

## 6. Conclusions

This study provides novel insights into how political alignment shapes neural and behavioral responses to contested public policies. By integrating EEG data with a naturalistic judgment task, we identified significant differences in gamma-band activity—particularly in the right prefrontal cortex—between individuals who supported and those who opposed a government-mandated COVID-19 vaccine policy. These differences are interpreted as markers of motivated attention, cognitive–affective integration, and value-based engagement. Crucially, our findings illustrate that behavioral divergence in attentional and evaluative engagement, observable through EEG spectral dynamics, emerges as a function of political identity. This study contributes to affective neuroscience and behavioral science by linking ideology-driven processing biases to measurable neurophysiological patterns during real-world policy evaluation. Such results not only clarify the mechanisms of political polarization but also underscore the utility of EEG in capturing implicit evaluative behaviors relevant to broader models of decision-making and public persuasion.

## Figures and Tables

**Figure 1 behavsci-15-01043-f001:**
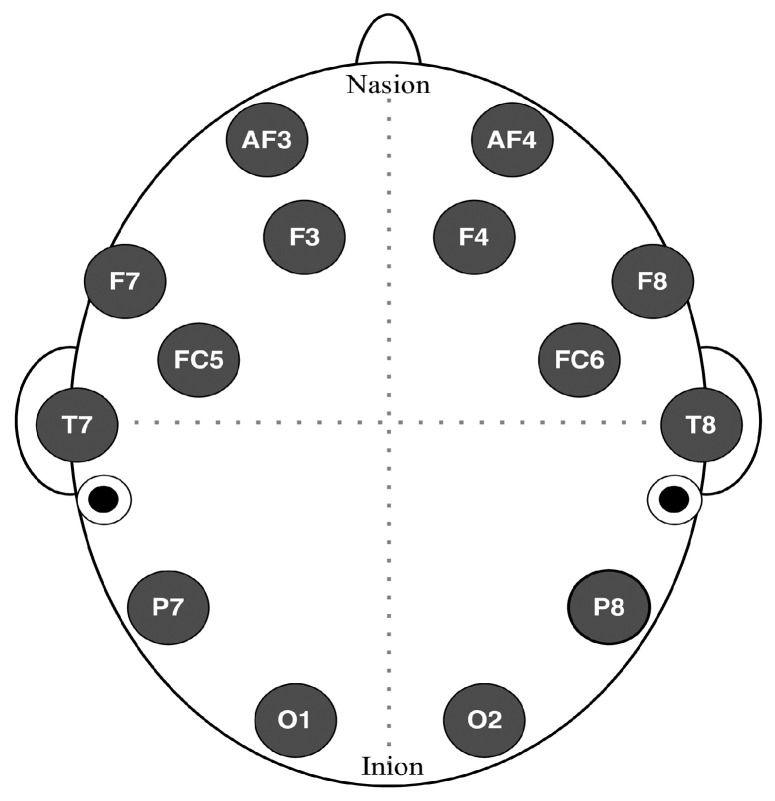
Electrode placement of the Emotiv EPOC+ device. Adapted from ([15]).

**Figure 2 behavsci-15-01043-f002:**
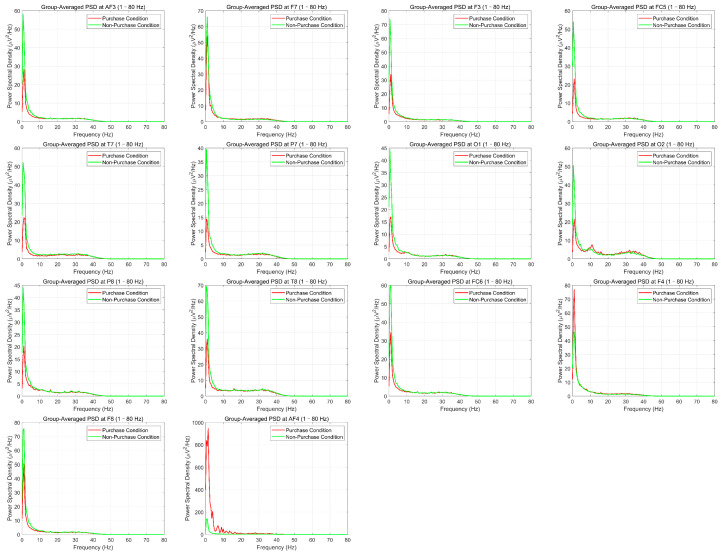
Grand-Averaged PSD Curves Across Channels for support vs. Non-support groups.

**Figure 3 behavsci-15-01043-f003:**
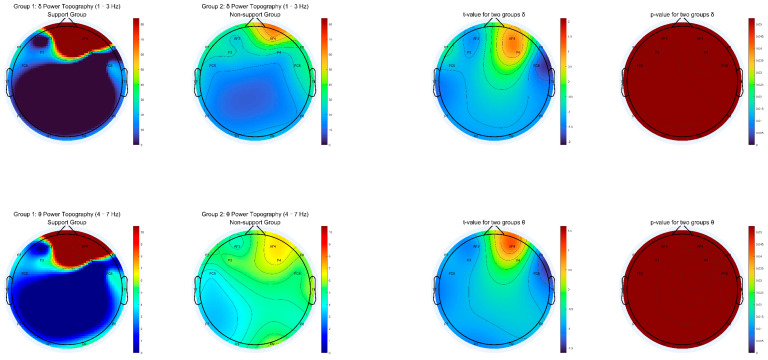
Topographic Distribution of Delta and Theta Power for Support vs. Non-support groups (Includes mean PSD, t-statistics, and FDR-corrected *p*-value maps for 13 Hz and 4–7 Hz bands).

**Figure 4 behavsci-15-01043-f004:**
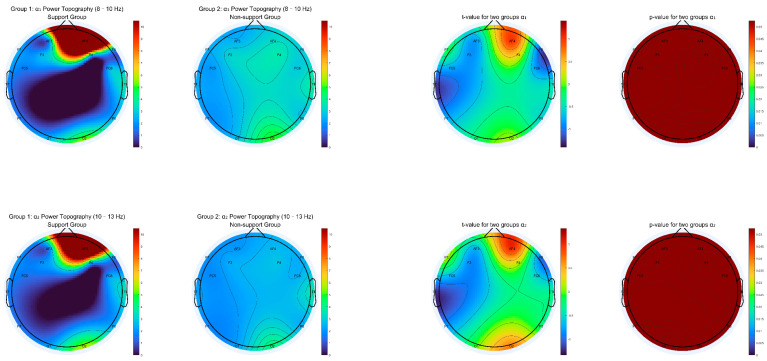
Topographic Distribution of Alpha1 and Alpha2 Power for Support vs. Non-support groups (Includes mean PSD, t-statistics, and FDR-corrected *p*-value maps for 8–10 Hz and 10–13 Hz bands).

**Figure 5 behavsci-15-01043-f005:**
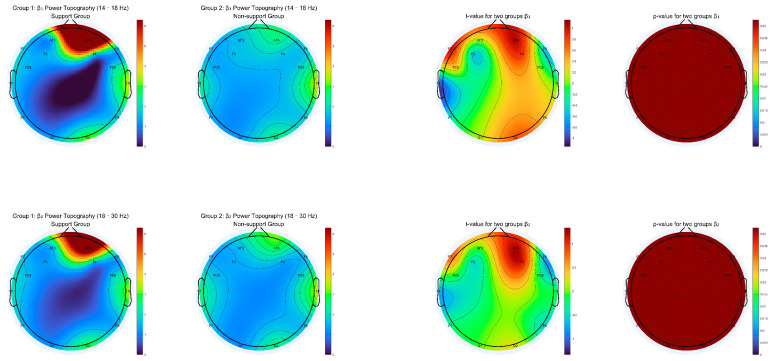
Topographic Distribution of Beta1 and Beta2 Power for Support vs. Non-support groups (Includes mean PSD, t-statistics, and FDR-corrected *p*-value maps for 14–18 Hz and 18–30 Hz bands).

**Figure 6 behavsci-15-01043-f006:**
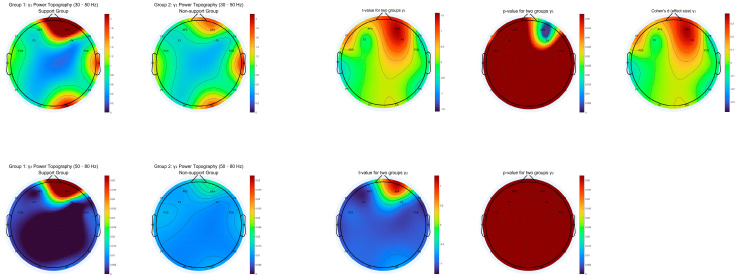
Topographic distribution of gamma1 and gamma2 power for support vs. non-support groups (Includes mean PSD, t-statistics, FDR-corrected *p*-value and Cohen’s d maps for 30–50 Hz and 50–80 Hz bands).

**Figure 7 behavsci-15-01043-f007:**
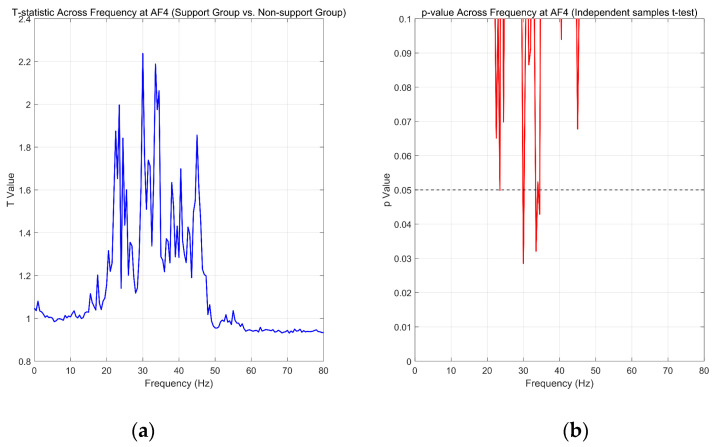
Frequency-resolved statistical comparison at AF4 reveals gamma1 band differences. (**a**): frequency-specific t-statistics. (**b**): corresponding *p*-values (right). Significant differences (*p* < 0.05) emerge in the gamma1 (30–50 Hz) bands, indicating elevated high-frequency activity in support group.

**Table 1 behavsci-15-01043-t001:** *t*-Test Results for Whole-Brain Averaged EEG Power Across Frequency Bands.

Band	*t*	*df*	*SD*	*p*	*CI* [Low]	*CI* [High]
delta	0.77	34	211.03	0.45	−45.12	99.87
theta	0.72	34	31.24	0.48	−6.92	14.55
alpha1	0.86	34	13.6	0.40	−2.69	6.65
alpha2	0.91	34	8.92	0.37	−1.7	4.43
beta1	1.09	34	4.39	0.28	−0.7	2.32
beta2	1.01	34	2.08	0.32	−0.36	1.07
gamma1	1.12	34	0.77	0.27	−0.12	0.41
gamma2	−0.17	34	0.05	0.87	−0.02	0.02

*Note.* Frequency bands were defined as follows: delta (1–4 Hz), theta (4–8 Hz), alpha1 (8–10 Hz), alpha2 (10–13 Hz), beta1 (13–18 Hz), beta2 (18–30 Hz), gamma1 (30–50 Hz), and gamma2 (50–80 Hz). For each participant, EEG power spectral density (PSD) values were averaged across all scalp channels within each band. Independent-samples t-tests were conducted to compare mean band power between the supporter and opponent groups. *SD* = pooled standard deviation; *CI* = 95% confidence interval.

## Data Availability

All datasets employed in this research are archived in the Science Data Bank and are available for private access at the following URL: https://www.scidb.cn/en/anonymous/cXVpUWIy (accessed on 15 May 2025).

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
