# Peer review of "Cognitive–Affective Dynamics of Political Attitude Polarization: EEG-Based Behavioral Evidence from a COVID-19 Vaccine Mandate Task"

_behavsci, 2025, doi:10.3390/bs15081043_

Round 1

Reviewer 1 Report

Comments and Suggestions for Authors

The study presents an interesting case for affective neuroscience and political psychology, by demonstrating that political attitudes modulate gamma-band EEG activity during policy evaluation. More specifically, power spectral density (PSD) analyses were performed to compare how much band-specific activity (exploratorily) participants exhibited while reading a vaccine mandate policy, under the assumption that higher gamma activity reflects greater cognitive or emotional engagement. Group differences in PSD, particularly in the right prefrontal cortex, were interpreted as neural signatures of political alignment.

The study design and analytical approach introduce a potentially novel contribution in the field, as no prior study in the context of political neuroscience has incorporated group-differences in information processing during a naturalistic (self-paced) reading task. However, this choice comes with certain methodological constrains, which beg attention before the study results can be confidently understood. Attached, I outline certain major and minor issues of concern, both at the theoretical and at the methodological level,  which, in my view, need to be addressed before accepting the study for further consideration.

Reviewer 2 Report

Comments and Suggestions for Authors

This manuscript addresses a socially and scientifically relevant topic - the neural underpinnings of political polarization in the context of vaccine mandates. The paper is promising and clearly situated within a growing interdisciplinary field. However, in its current form, several foundational issues require attention before the manuscript can reach its full potential.

Line-by-Line Comments

  • Abstract: Needs stronger contextualisation and justification of the study’s contribution. Reduce methodological detail and revise the final sentence to briefly indicate implications.
  • Lines 28–34: This content should be paraphrased and incorporated into the abstract.
  • Lines 28–29: Lacks sources. If this is a longstanding question, cite foundational or recent literature.
  • Lines 29–31: Unsupported claim. Who has argued this? Needs referencing.
  • Lines 37–39: Echo chamber claims should be updated with more recent and robust sources.
  • Lines 39–42: The Kitamura and Ihara (2023) citation links to a preprint. It is not listed in the references and lacks peer-review status. If retained, this must be explicitly justified.
  • Lines 45–48: These sources are outdated given the contemporary nature of the topic. Supplement with more recent research.
  • Line 48: Schuitema et al. (2010) is cited to support norm-based evaluation claims, but the article focuses on a specific case (Stockholm congestion charge) and does not generalise cognitive mechanisms. Broaden the theoretical base or qualify the claim.
  • Line 50: Jones et al. (2011) is a theoretical synthesis, not an empirical study. Rework the sentence or replace with empirical references.
  • Lines 49–50: McBeth et al. (2005) does not empirically support claims about fairness, risk, or benefit. Use appropriate psychological sources.
  • Line 74–75: Missing citation.
  • Lines 75–78: Claim is underdeveloped. Add more sources—especially recent ones—to support the neuroscience claims.
  • Lines 84–87: Bold claims require more extensive referencing.
  • Lines 87–89: “Says who?” This needs substantiation.
  • Line 91: Author name and reference are in all caps—should be corrected. As with other older sources, complement with newer supporting studies.
  • Line 118–120: Strong claim without a source. Provide evidence or rephrase to be less absolute.
  • Lines 120–122: Clarify which studies are being referenced.
  • Lines 124–126: Two sources (both on commercials) are insufficient to generalise about EEG's relevance for political analysis. Add breadth.
  • Lines 130–132: Reword. One source does not allow for general conclusions. At best, it suggests a trend.
  • Lines 135–136: Two sources for a broad claim are not enough.
  • Lines 138–139: The conclusion is offered before the data. Reorganise to show evidence first.
  • Lines 144–147: The link between commercial and political EEG applications is unconvincing. Needs stronger support.
  • Lines 151–158: Largely repeats earlier content. Streamline. Again, start from evidence and build to hypothesis.
  • Lines 162–165: Overgeneralisation. Most countries were compliant with COVID-19 vaccination. Specify context or rethink this framing.
  • Lines 165–168: Similar issue—be specific. Without geographical focus, generalising global sentiment is problematic.
  • Lines 194–197: H1 is plausible but vague. Clarify EEG power specifics (frequency bands, topography). Avoid overinterpreting EEG as indicative of engagement without supporting data.
  • Lines 198–201: H2 goes further but includes imprecise language ("direction of neural activity"). Rephrase and support with relevant literature. Clarify frequency bands and cognitive-affective mappings.
  • Lines 205–227: Specify participant age range.
  • Line 233: Which AI was used to generate the stimulus, specify the model and the prompt used to generate the documents. Clarify if documents were human reviewed. Justify this methodological decision and consider the interpretative risks. Also address why not use real policy documents? Also what the policy documents specifically said? What was the main theme?

Round 2

Reviewer 2 Report

Comments and Suggestions for Authors

After the revisions the paper is now ready to be published.